# Modeling Pedestrian Motion in Crowded Scenes Based on the Shortest Path Principle

**Yi Zou and Yuncai Liu \***

Key Laboratory of System Control and Information Processing, Ministry of Education of China, Department of Automation, Shanghai Jiao Tong University, Shanghai 200240, China; zouyimumu@gmail.com
\* Correspondence: whomliu@sjtu.edu.cn

**Abstract:** In the computer vision field, understanding human dynamics is not only a great challenge but also very meaningful work, which plays an indispensable role in public safety. Despite the complexity of human dynamics, physicists have found that pedestrian motion in a crowd is governed by some internal rules, which can be formulated as a motion model, and an effective model is of great importance for understanding and reconstructing human dynamics in various scenes. In this paper, we revisit the related research in social psychology and propose a two-part motion model based on the shortest path principle. One part of the model seeks the origin and destination of a pedestrian, and the other part generates the movement path of the pedestrian. With the proposed motion model, we simulated the movement behavior of pedestrians and classified them into various patterns. We next reconstructed the crowd motions in a real-world scene. In addition, to evaluate the effectiveness of the model in crowd motion simulations, we created a new indicator to quantitatively measure the correlation between two groups of crowd motion trajectories. The experimental results show that our motion model outperformed the state-of-the-art model in the above applications.

**Keywords:** crowd motion; computer vision; motion model of pedestrians; shortest path principle; origin and destination

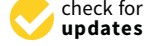

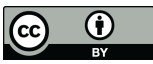

## 1. Introduction

Crowd motion is a common phenomenon in human society, which often appears in train stations, shopping malls, street intersections and other mass events. As shown in Figure 1, a crowd usually includes a large number of individuals, and some collective patterns appear when the individuals are moving. To another point, gatherings of so many people bring potential risks to public safety, such as stampedes, terrorist attacks and so on [1,2]. In recent years, researchers analyzed crowd motions with computer vision technology to extract real-time information for city emergency managers [3–5]. For instance, crowd collectiveness was proposed as a quantitative indicator to measure the degree of orderliness of crowd motions [6,7]. From another perspective, abnormal behaviors were detected from crowd motions to reveal the potential hazards for social security [8,9]. Moreover, some studies estimated the density and distribution of crowd motions to provide guidelines for designing public places [10,11].

Describing the motions in crowded scenes is one of the core issues of crowd motion analysis in computer vision-related fields. Tracking individuals and learning motion patterns are the two traditional topics, which describe the crowd motions at the microscopic and macroscopic levels, respectively. Tracking individuals in a crowd provides an intuitive and detailed description of crowd motion. However, due to technical difficulties, tracking individuals in dense crowds still remains a relatively unexplored problem [12]. Thus, following motion patterns gives a relatively reliable result for describing crowd motion. However, instead of the entire moving process, motion patterns just describe the rough motion tendency of individuals. To combine the advantages of the above two strategies, Zhou et al. proposed a mixture model of dynamic pedestrian agents (MDA) to build the

motion models of pedestrians in a crowd [13]. Motion models of pedestrians have many significant applications. For example, given a motion model, a crowd's motion can be described with only a few parameters. Consequently, we can simulate the pedestrians' motions with partially observed trajectories. Additionally, the motion models can be applied to traffic flow statistics, crowd motion simulation and so on.

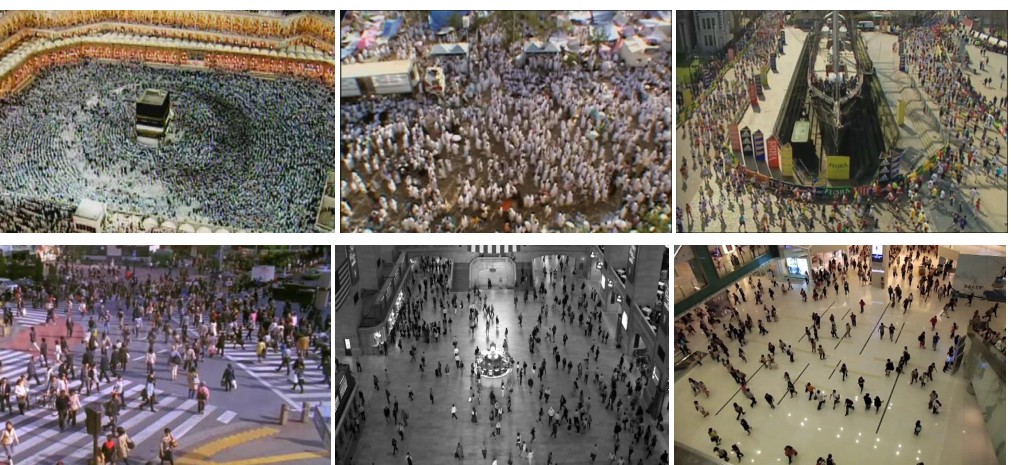

**Figure 1.** Examples of crowd scenes. A crowd usually includes a large number of individuals.

MDA is an agent-based model that treats the pedestrians as dynamic pedestrian-agents and uses the affine transformation matrix as the control rule to drive the agent. However, a single matrix can not accurately model the complicated movements of a pedestrian. In this work, we revisit the related work in social psychology to summarize the rules governing pedestrian motions. Sociologists indicated that pedestrians always take the shortest possible paths to reach a certain destination, and we call this "shortest path principle." In crowd motions, the collective wills of pedestrians reinforce the shortest path principle. The motion of a pedestrian consists of three basic elements: origin, destination, and the movement path from the origin to the destination. Based on the shortest path principle, we propose a two-part motion model to describe these basic elements of pedestrian motions. One part estimates the probability of a pedestrian reaching each candidate pair of origin and destination, and the other generates the movement path of a pedestrian between any two points in the scene. We use the trajectories of feature points obtained by the Kanade–Lucas–Tomasi (KLT) tracker as the observations of pedestrian motions.

In this paper, we first build a shortest path graph according to the trajectories of feature points. Based on the shortest path principle, this graph can be used to generate the movement path of a pedestrian between any two points in the scene. Then, we quantify the shortest path principle as the shortest path likelihood. Combined with the connectivity between trajectories, we propose a generalized path likelihood to estimate the probability of the trajectory reaching each candidate pair of origin and destination. Finally, we identify the origin and destination of the trajectory according to the generalized path likelihood. In the experiments, we verified the effectiveness of the proposed motion model in the following applications: a pedestrian motion simulation, motion pattern classification and traffic flow statistics, and crowd motion description and simulation. In addition, we compared the proposed model with the MDA model on the New York Grand Central Station dataset. Experimental results show that our motion model outperformed the MDA model in the above applications.

We summarize the innovations and contributions of this work as follows. (1) We propose a two-part motion model to directly describe the basic elements of pedestrian motion. The model is able to describe human moving behavior in a crowd in effective and flexible ways. (2) Based on the sociological findings, we integrated the shortest path principle into the proposed motion model, making our motion model highly consistent with the behavioral decision-making of pedestrians. (3) We combined the shortest path

principle with the connectivity between trajectories to propose a likelihood framework to find the origin and destination of a trajectory. This strategy takes advantage of the spatial and temporal information of the trajectories. (4) To assess the simulation of crowding behavior, we propose an indicator to quantitatively measure the correlation between two groups of crowd motion trajectories.

## 2. Related Work

Crowd motion description is an interdisciplinary subject, building mainly on social psychology and computer vision. In the social psychology field, a lot of fundamental jobs have been done [14], which provide guidelines for scholars in other research fields. Helbing and Molnár pointed out that people want to reach their destinations as comfortably as possible, which means that a pedestrian usually takes the shortest possible path [15]. Le Bon showed that when pedestrians are gathered in a crowd, their conscious personalities vanish while the common desire is amplified [16]. This constitutes the theoretical basis of the core idea of this paper: the shortest path principle. In the computer vision field, individual tracking, motion pattern learning and the agent-based model are the most representative topics on crowd motion description. We attempted to combine the research results from these two disciplines to propose a new framework to describe crowd motions.

Tracking individuals is a very intuitive task, which provides elaborated and flexible results to describe crowd motion. Ali and Shah first proposed a method based on floor field to track individuals in high-density structured scenes [17]; then Rodriguez et al. employed a correlated topic model to promote the job in unstructured scenes [18]. Ge et al. detected and tracked individuals utilizing a reversible-jump Markov chain and Monte Carlo model [19]. Kratz and Nishino, and Zhang et al. applied, respectively, the local Spatio-temporal motion patterns and local patch motion patterns to track individuals in extremely crowded scenes [20,21]. Bera and Manocha combined discrete and continuous flow models to extract the trajectory of each pedestrian in moderately dense crowd videos [22]. However, tracking results for individuals are not reliable due to the issue of frequent blocking in crowded scenes. To solve this problem, Lin et al. proposed a tube-and-droplet-based approach to represent motion trajectories [23]. Ren et al. have performed a lot of studies on optical flow to improve the reliability of feature point tracking [24–26].

Learning motion patterns in a crowd is another hot point, which provides relatively reliable results for describing the crowd motion. Hu et al. extracted super tracks to represent the corresponding motion patterns based on the global motion field [27]. Saleemi et al. built a statistical representation of motion patterns with the Gaussian mixture model [28], and Mehran et al. adopted a streakline representation of flow to segment motion patterns in crowd [29]. Wang et al. and Zhou et al. used topic models to build generative models for motion patterns [30,31]. Wang et al. analyzed motion patterns by clustering the hybrid generative-discriminative feature maps [32]. Tokmakov et al. learned motion patterns in videos with the aid of a fully convolutional network [33]. Additionally, many studies have been performed to learn motion patterns in crowds by clustering trajectories of feature points [34–38]. However, motion patterns only describe the rough motion tendencies of individuals instead of the entire trajectories.

The agent-based model is a powerful tool with which to describe crowd motion. Ali and Shah formulated the individuals in the crowd as cellular automaton agents driven by a scene structure-based force model [17], and Hu et al. expressed the motion patterns as super track agents moving with the motion flow field [27]. The driving rules of the above agents change with space and time; in other words, they are local rules. Zhou et al. proposed a method called the mixture model of dynamic pedestrian-agents (MDA), which treated the pedestrians as dynamic pedestrian-agents, and the transition processes of dynamic agents were formulated as affine transformation matrices [13]. The innovation of Zhou's work is that they used a global rule (a fixed affine transformation matrix) to govern an agent throughout the entire transition process. However, a fixed matrix is too simple to model the complicated moving process of a pedestrian. Inspired by the MDA model, we propose a new model based on the shortest path principle to describe pedestrian motion in a crowd.

## 3. Model Learning

In this paper, the shortest path principle is treated as the control rule for pedestrian motions. Under this rule, we propose a new motion model, which consists of two parts. One part is the shortest path graph, which generates the movement path of a pedestrian between any two points in the scene. The other is the generalized path likelihood, which estimates the probability of a pedestrian reaching each candidate pair of origin and destination. In this section, we first build the shortest path graph. Then, we quantify the shortest path principle as a likelihood and define the connectivity between trajectories. Finally, we combine the shortest path likelihood with the connectivity between trajectories to propose a generalized path likelihood.

### 3.1. Data Pre-Processing

We divided a scene image into non-overlapping equal-sized grids with $N \times N$ pixels, as shown in Figure 2b. Each grid corresponds to a node of the shortest path graph. We labeled the entrance/exit regions of the scene by hand, assuming that the pedestrians can only enter or leave the scene from these regions. The grids in entrance/exit regions are the candidate origins and destinations, as shown in Figure 2c. We used the Douglas Peucker algorithm to simplify the original trajectories to poly-lines and segment the trajectories according to the turning points. As shown in Figure 3, the original trajectory *AD* was divided into three new trajectory segments, *AB*, *BC* and *CD*, which can be approximated as straight lines. We used the trajectory segments instead of the original trajectories as the observations for the motion model.

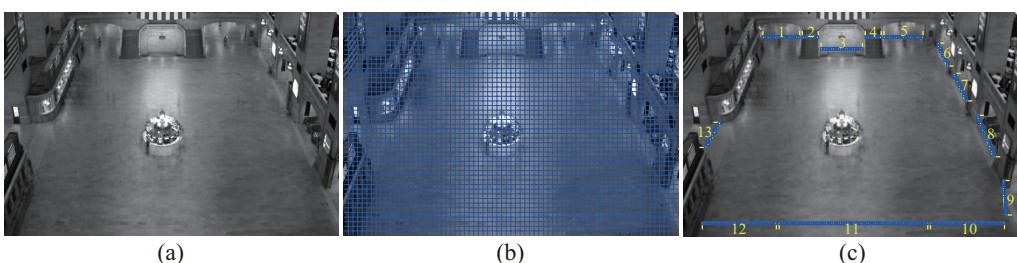

| (a) | (b) | (c) |

**Figure 2.** (**a**) The original scene image. (**b**) The non-overlapping grids. (**c**) The human-labeled entrance/exit regions. Blue grids are the candidate origins and destinations.

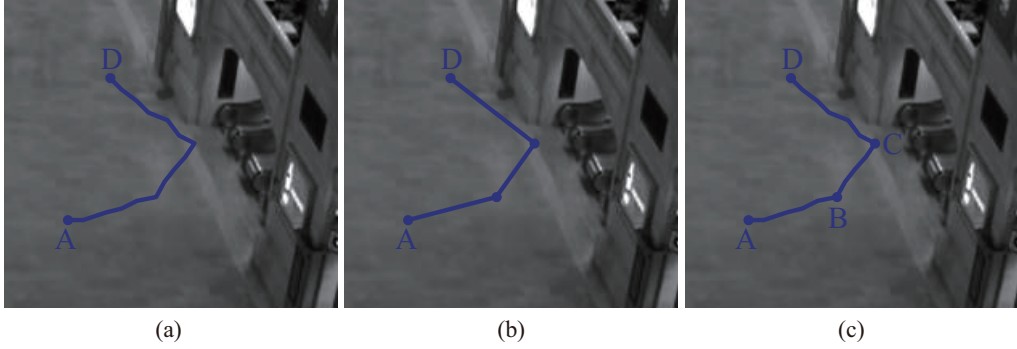

| (a) | (b) | (c) |

**Figure 3.** (**a**) The original trajectory *AD*. (**b**) The simplified polyline obtained by the Douglas Peucker algorithm. (**c**) The three new trajectory segments *AB*, *BC* and *CD*.

### 3.2. Shortest Path Graph

Although the shortest possible paths is preferred for reaching one's destination, pedestrians usually do not walk in straight lines due to obstacles and human limitations. The shortest path graph was built on the grids of the scene image to generate the movement path of a pedestrian between any two grids in the scene. We recorded the distances between pairs of grids in the shortest path graph. Initially, the distances between all grids were set to infinity. Then the adjacent grids through which the trajectory passed were identified,

and the distances between them were updated. As shown in Figure 4a, a trajectory passed through the grids *A*1, *B*1, *B*2, *C*2, *C*3, *D*3 and *E*4. The distances of grid *A*1 to grid *B*1, grid *B*1 to grid *B*2, grid *B*2 to grid *C*2, grid *C*2 to grid *C*3 and grid *C*3 to grid *D*3 were recorded as 1. The distance of grid *D*3 to grid *E*4 was recorded as $\sqrt{2}$. Next, we updated the distance between the start grid and the end grid of the trajectory. Based on the poly-lines simplification and segmentation, the trajectory could be approximated as a straight line. As shown in Figure 4b, we recorded the distance of grid *A*1 to grid *E*4 as 5. The calculation of the shortest path between any two grids can be implemented with the dynamic programming algorithm, and it is treated as the movement path of a pedestrian.

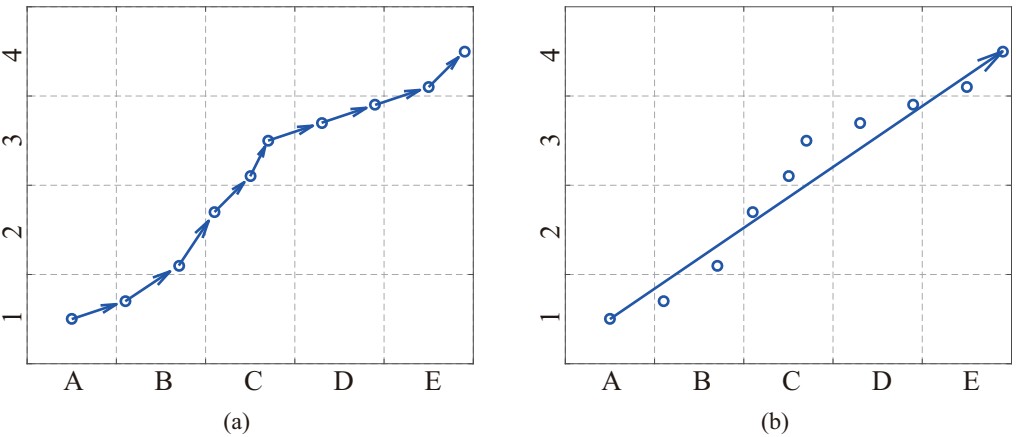

(a)  (b)

**Figure 4.** A simple example to show the distance recording process in the shortest path graph. (**a**) Recording the distance between the adjacent grids through which the trajectory passes. (**b**) Recording the distance between the start grid and the end grid of the trajectory.

### 3.3. Shortest Path Likelihood

To measure the probability of a trajectory reaching each candidate pair of origin and destination, we quantified the shortest path principle as a likelihood in a straightway. As shown in Figure 5a, the grids $g_a$ and $g_b$ are the start grid and end grid of the trajectory *ab*, and the grids $g_o$ and $g_d$ are the candidate origin and destination grids. We define the shortest path likelihood of the trajectory *ab* reaching the origin $g_o$ and destination $g_d$ as follows:

$$L_{ab}(g_o, g_d) = \frac{S(g_o, g_d) - S(g_o, g_a) - S(g_b, g_d)}{S(g_o, g_a) + S_{tr}(g_a, g_b) + S(g_b, g_d)}. \tag{1}$$

$S(g_i, g_j)$ is the shortest distance from grid $g_i$ to grid $g_j$ calculated by the shortest path graph; $S_{tr}(g_a, g_b)$ is the shortest distance from grid $g_a$ to grid $g_b$ and through trajectory *ab*. $S(g_o, g_d)$ represents the length of an ideal shortest path from origin $g_o$ to destination $g_d$; $S(g_o, g_d) - S(g_o, g_a) - S(g_b, g_d)$ represents the effective projection length of the trajectory *ab* on the ideal shortest path; and $S(g_o, g_a) + S_{tr}(g_a, g_b) + S(g_b, g_d)$ represents the length of a feasible shortest path of trajectory *ab* from origin $g_o$ to destination $g_d$.

We use the ratio between the effective projection length of the trajectory and the length of the feasible shortest path to construct the shortest path likelihood. A larger numerator means a greater intersection between the trajectory and the ideal shortest path, and a smaller denominator means more consistency between the feasible shortest path and the ideal shortest path. However, the effective projection lengths of the trajectories are usually very small because of the short trajectories, which makes the shortest path likelihood unreliable. Targeting this problem, we combine the connectivity between trajectories to upgrade the shortest path likelihood.

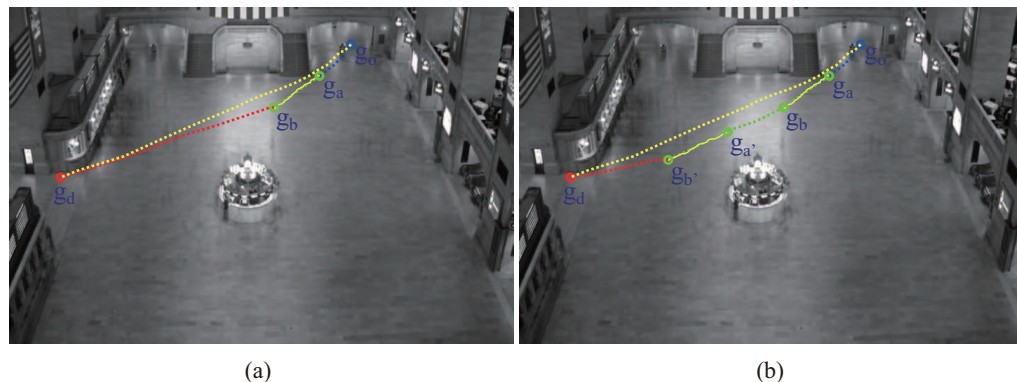

(a)               (b)

**Figure 5.** (**a**) An example to explain the definition of shortest path likelihood. The yellow line is the trajectory *ab*; the small circles represent the grids of the start point, end point, candidate origin and destination of the trajectory *ab*. The dashed lines represent the shortest paths between two grids generated by the shortest path graph. The yellow dashed line is the ideal shortest path from origin $g_o$ to destination $g_d$. The blue dashed line, the green dashed line and the red dashed line combine into the feasible shortest path of trajectory *ab* from origin $g_o$ to destination $g_d$. (**b**) An example to explain the definition of generalized path likelihood. The yellow lines represent the trajectory *ab* and trajectory $a'b'$; the green dashed line represents the approximate trajectory $ab'$ of the generalized trajectory $\overline{ab'}$. The representations of other symbols are the same as in (**a**).

### 3.4. Connectivity between Trajectories

In crowd scenes, pedestrians are often occluded by each other. When an occlusion occurs, the old feature point will stop tracking, and a new feature point will be detected and tracked. The trajectories tracked for the same pedestrian at different times should be connected. However, we cannot explicitly determine which trajectories belong to the same pedestrian. Therefore, we define the connectivity between trajectories to indicate whether there is a possible connection between two trajectories. Since the motion of pedestrians is continuous in space and time, the connectivity between trajectories depends on the travel times and relative positions of trajectories.

Suppose that there are trajectories *ab* and $a'b'$, and trajectory *ab* is before trajectory $a'b'$ in time. We define that there is a possible connection between trajectories *ab* and $a'b'$ if they satisfy the following two conditions:

(1)    $t_e^{ab} \leq t_s^{a'b'} \leq t_e^{ab} + T$,

(2)    $d\left( X_e^{ab} + v_{ab} \cdot \left( t_s^{a'b'} - t_e^{ab} \right), X_s^{a'b'} \right) \leq D$ or

      $d\left( X_e^{ab}, X_s^{a'b'} - v_{a'b'} \cdot \left( t_s^{a'b'} - t_e^{ab} \right) \right) \leq D$.

Here, $t_e^{ab}$ represents the end time of trajectory *ab*, $t_s^{a'b'}$ represents the start time of trajectory $a'b'$ and *T* is the threshold constant of time interval. $X_e^{ab}$ represents the end position of trajectory *ab*; $X_s^{a'b'}$ represents the start position of trajectory $a'b'$; $v_{ab}$ and $v_{a'b'}$ represent the velocities of trajectories *ab* and $a'b'$; $d\left( X_i, X_j \right)$ represents the distance between position *i* and position *j*; and *D* is the threshold constant of space interval. Conditions (1) and (2) guarantee the continuity of the possible trajectory connection in time and space, respectively, but they are not sufficient for the connection of trajectories.

### 3.5. Generalized Path Likelihood

We combine the shortest path likelihood with the connectivity between trajectories to propose a generalized path likelihood. As above, we take the trajectory *ab* as an example. As shown in Figure 5b, we suppose that there is a possible connection between the trajectories *ab* and $a'b'$, and trajectory *ab* is in front of trajectory $a'b'$ in time. We connect trajectory *ab* with trajectory $a'b'$ to construct a generalized trajectory $\overline{ab'}$. Since the trajectory $\overline{ab'}$ does not practically exist, we generate an approximate trajectory $ab'$ to represent the generalized trajectory $\overline{ab'}$ based on the shortest path principle. The trajectory $ab'$ is composed of

trajectory $ab$, trajectory $a'b'$ and the shortest path between the points $b$ and $a'$. We defined the shortest path likelihood of the generalized trajectory $\overline{ab'}$ reaching the origin $g_o$ and destination $g_d$ as:

$$L_{\overline{ab'}}(g_o, g_d) = P_{\overline{ab'}} \cdot L_{ab'}(g_o, g_d). \tag{2}$$

where,

$$P_{\overline{ab'}} = \frac{S(g_o, g_{b'}) - S(g_o, g_a)}{S_{tr}(g_a, g_{b'})},$$

$$L_{ab'}(g_o, g_d) = \frac{S(g_o, g_d) - S(g_o, g_a) - S(g_{b'}, g_d)}{S(g_o, g_a) + S_{tr}(g_a, g_{b'}) + S(g_{b'}, g_d)},$$

$$S_{tr}(g_a, g_{b'}) = S_{tr}(g_a, g_b) + S(g_b, g_{a'}) + S_{tr}(g_{a'}, g_{b'}).$$

Here, $P_{\overline{ab'}}$ represents the probability of the existence of generalized trajectory $\overline{ab'}$. $S_{tr}(g_a, g_{b'})$ represents the length of a feasible shortest path of generalized trajectory $\overline{ab'}$, $S(g_o, g_{b'}) - S(g_o, g_a)$ represents the effective projection length of generalized trajectory $\overline{ab'}$ on the ideal shortest path from origin $g_o$ to grid $g_{b'}$. We use the ratio between them to measure the probability of the existence of generalized trajectory $\overline{ab'}$. $L_{ab'}(g_o, g_d)$ is the shortest path likelihood of the trajectory $ab'$. Then, we replace the trajectory $ab$ with the generalized trajectory $\overline{ab'}$ and connect the trajectory with another trajectory to construct a new generalized trajectory. We update the shortest path likelihood of the new generalized trajectory and then iterate the above process until the likelihood is no longer increased. We start with trajectory $ab$, and use $A^*$ algorithm to search the generalized trajectory, which has the maximum shortest path likelihood in all possible trajectory connections, and take the shortest path likelihood of this generalized trajectory as the generalized path likelihood of the trajectory $ab$ reaching the origin $g_o$ and destination $g_d$. The generalized path likelihood makes full use of the spatial and temporal information of the trajectories, which makes it more reliable.

## 4. Origin and Destination Seeking

Seeking the origin and destination of a trajectory is the core work and the basis of the experiments in this paper. We label the entrance/exit regions in the scene, and find the candidate grids of the origins and destinations, as shown in Figure 2c. We assume that a pedestrian enters and leaves the scene from the different entrance/exit regions. Therefore, the candidate origin and destination of a trajectory should be selected from different entrance/exit regions. First, we pick two entrance/exit regions as the entrance and exit for the trajectory. We take the grids in the entrance as the candidate origins and the grids in the exit as the candidate destinations. We then calculate the generalized path likelihood of the trajectory reaching each candidate pair of origin and destination, and select the pair with the maximum likelihood as the potential pair of origin and destination of the trajectory for this combination of entrance and exit. Then, we traverse all combinations of entrance and exit and find the corresponding potential pairs of origin and destination. Among all of the potential pairs, the one with the largest likelihood is chosen as the final origin and destination of the trajectory.

## 5. Experiment Results Analysis

We compare the proposed motion model with the MDA model, which is the best motion model to our knowledge. The surveillance video of New York Central Station was chosen as the experimental object. All experimental results are reported for a 30-min video with a frame rate of 24 frames per second and a resolution of $480 \times 720$. The size of image grids was set to $5 \times 5$ pixels. We extracted and tracked the feature points using the KLT tracker to obtain the original trajectories, and trimmed the trajectories beyond the entrance/exit regions. The original trajectories were simplified and segmented by the Douglas Peucker algorithm, and the threshold of the algorithm was set to $\frac{5}{2}\sqrt{2}$ pixels. In all the experiments in this paper, we used the trajectories processed above as the observations of the pedestrian motions, as shown in Figure 6a. The number of trajectories was about

160,000. The length histogram in Figure 6b shows that most of the trajectories were very short.

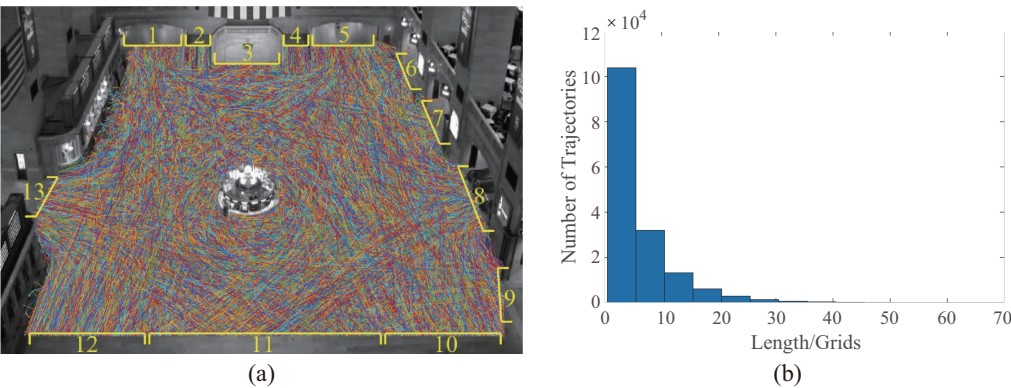

(a)                                   (b)

**Figure 6.** A brief introduction of the experimental data. (**a**) We show the trajectories with different colored lines, and the human labeled entry/exit regions by yellow bounding boxes. (**b**) A histogram of the trajectory lengths; the unit of length is the number of image grids. Most of the trajectories are very short.

### 5.1. Pedestrian Motion Simulation

The objective of this experiment was to simulate the pedestrian motions by the partially observed trajectories. Here, we take the movement path to represent the pedestrian motion. Other simulation details, such as speed, will be discussed in the crowd motion simulation experiment below. We found 680 complete trajectories in the original trajectories obtained by the KLT tracker, which reached the entrance and exit regions in the scene, as shown in Figure 7a, and they were taken as the true movement paths of pedestrians. We simplified and segmented the complete trajectories with the Douglas Peucker algorithm, and obtained 2428 trajectory segments. These segments were taken as the partially observed trajectories, and each trajectory corresponds to an independent pedestrian. Given a partially observed trajectory, we first found the origin and destination of it. Then, we generated the feasible shortest path of the trajectory from the origin to destination as the simulated movement path of the pedestrian. The MDA model uses an affine transformation matrix to expand the past and future movement paths of the trajectory to generate the simulated movement path. We use both the pattern accuracy of the simulated paths and the distance between the simulated and the true path to evaluate the simulation.

First, we analyzed the pattern accuracy of the simulated paths. We judged the pattern of a simulated path to be correct if the entrance and exit of the simulated path were the same as those of the corresponding true path. We show the pattern accuracies of the simulated paths obtained by our model and the MDA model in Figure 7b,c, respectively. The accuracy of our model was 74.92%; that of the MDA model was 53.42%. The green trajectory indicates the correct simulated path, and the red trajectory indicates an erroneous one. It is worth noting that if the entrances and the exits of the trajectories are adjacent, the error rates of both models will increase markedly, which is demonstrated well by the dense red trajectories among the entrance/exit regions 8, 9 and 10. The dense regions exist because the regions near the entrance/exit are usually crowded; pedestrians are often forced to change their routes frequently when passing between an adjacent entrance and exit, causing difficulties for motion models. However, we can see that even in these extreme cases, the accuracy of our model was significantly higher than that of the MDA model.

Next, we calculated the distance between the simulated path and the true path. For a fair comparison, we only used the paths simulated correctly by both two motion models. We respectively sampled the simulated paths and the true paths at equal distances. The sampling frequency of the true paths was set to five times that of the simulated paths. For each sampling point on the simulated path, we took the minimum distance between it and all sampling points on the true path as the distance from the sampling point to the true path. The distance

between the simulated path and the true path was defined as the average distance from all sampling points on the simulated path to the true path. We calculated the mean values and standard deviations of the distances between the simulated paths and the corresponding true paths obtained by our model and the MDA model. In our results, the mean distance was 7.07; the standard deviation was 3.82. For the MDA model, the mean distance was 6.94; the standard deviation was 4.74. The unit of distance is the pixel. The mean distances obtained by the two models are approximately the same, though our model slightly outperformed the MDA model in terms of standard deviation. We show some examples of the simulated paths obtained by the two motion models in Figure 8 and mark the distances between the simulated paths and the true paths. Overall, on the premise of finding the correct entrance and exit, both models can simulate well the movement paths of pedestrians. However, if there are big turning points on the movement path, maybe caused by sudden direction changing or error tracking, the difference between the simulated path and the true path will increase significantly in both model results, just like the last two examples.

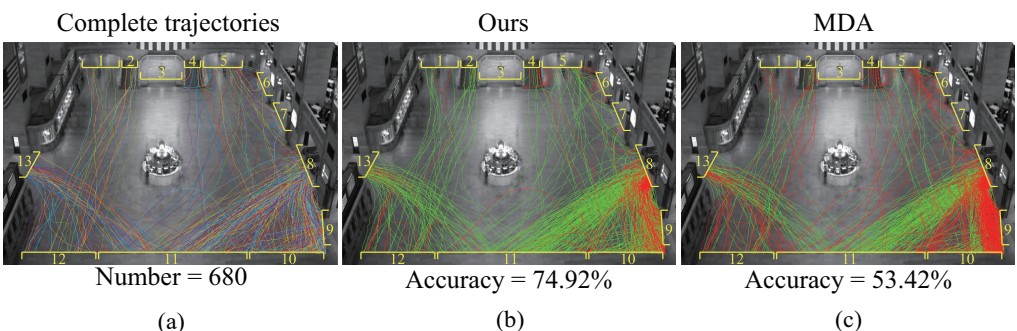

**Figure 7.** (**a**) The complete trajectories which reach the entrance and exit regions. The pattern accuracies of simulated paths obtained by our model (**b**) and the MDA model (**c**). The green trajectory represents a correct pattern of the whose simulated path, and the red trajectory represents an error.

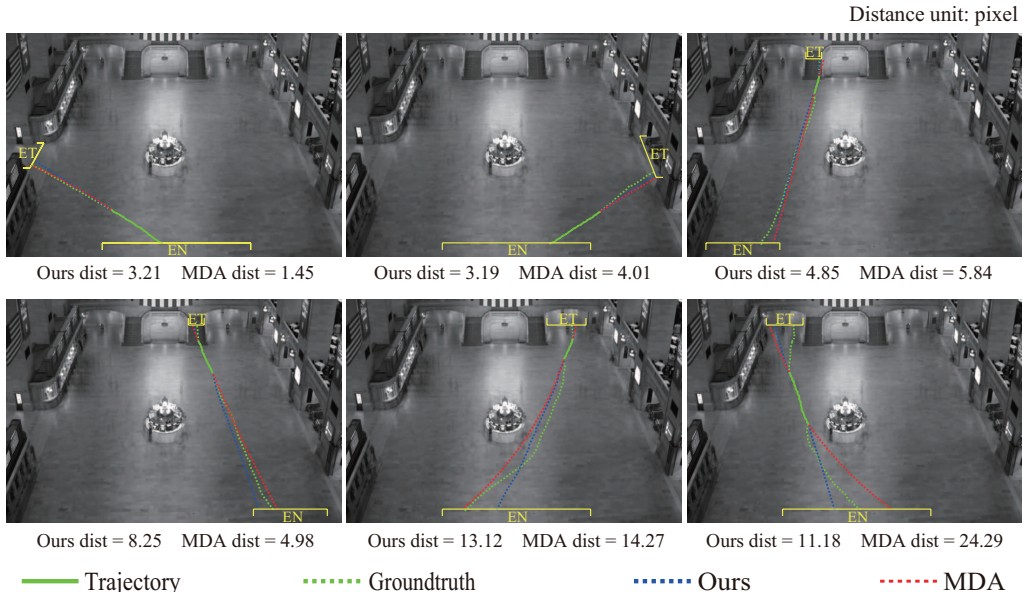

**Figure 8.** Some examples of the simulated paths obtained by our model and the MDA model. The green line represents the partially observed trajectory; the green dashed line represents the true path. The blue and red dashed lines represent the simulated paths obtained by our model and the MDA model, respectively. The EN and ET indicate the entrance and exit.

### 5.2. Motion Pattern Classification and Traffic Flow Statistics

We classified the motion pattern based on the entrances and exits of the scene. First, we categorized the trajectories whose origins and destinations belonged to the same entry

and exit regions into the same motion patterns. Then, we calculated the flow value of the trajectory according to the motion pattern to which it belonged. Taking the trajectory *ab* in Figure 5a as an example, it belongs to the motion pattern characterized by entrance 5 to exit 13. We define the flow value of trajectory *ab* as:

$$Fv_{ab} = \frac{S_{tr}(g_a, g_b)}{S(g_o, g_a) + S_{tr}(g_a, g_b) + S(g_b, g_d)}.$$

Finally, we summed up the flow values of all trajectories which belonged to a motion pattern to count the traffic flow of the motion pattern.

We show the top 12 motion patterns with the largest flow values obtained by our model and the MDA model in Figure 9a,b, and mark their flow values. We can see that there are 11 intersections between the 12 motion patterns obtained by these two models, which demonstrates that both models could identify the main motion patterns well in the scene. However, there are some differences in the spatial distributions of trajectories for the same motion patterns obtained by the two motion models, which is particularly obvious in the motion pattern of entrance 11 to exit 1. In the results of the MDA model, almost all trajectories in this motion pattern enter the scene from the right half of entrance 11 and pass along the left side of the central ticket machines. In our model's results, the trajectories enter the scene from every part of entrance 11, and a small portion of the trajectories pass along the right side of central ticket machines. This difference exists because the MDA model uses a linear dynamic system to constrain the shapes of motion patterns. All trajectories in a motion pattern approximately satisfy the same affine transformation, which limits the path selection of trajectory. Our motion model does not preset the shapes of the motion patterns but directly looks for the origin and destination of a trajectory based on the shortest path principle, which makes the trajectory free to select the path. To sum up, compared with the results of the MDA model, our results better retain the diversity of path decision-making caused by pedestrians' behavior habits and senses, surrounding environments and other uncertain factors in the real situation. Note that feature point extraction has limitations. For example, some pedestrians have several feature points, whereas others have no feature points. The trajectory flow value calculated in this paper can only be a rough measurement but not an accurate pedestrian counter.

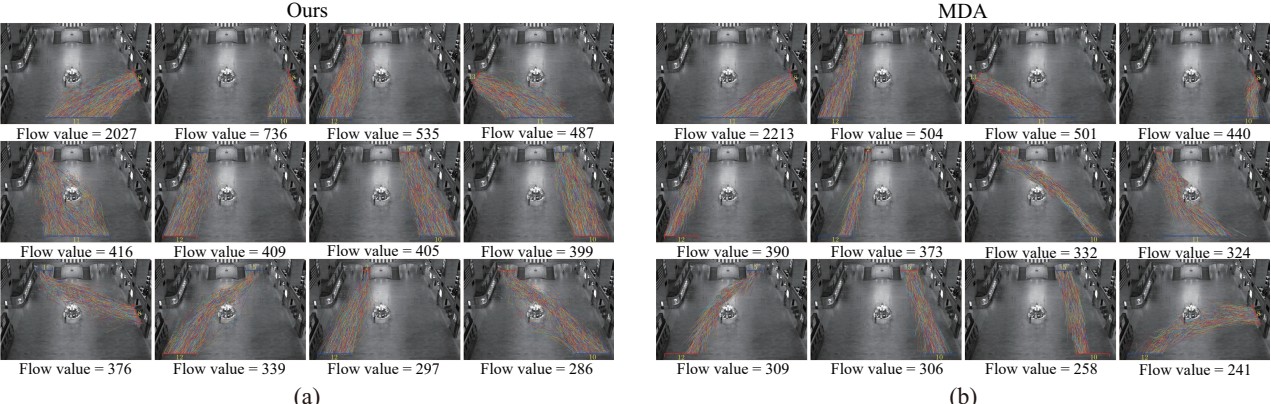

**Figure 9.** Top 12 motion patterns with the largest flow values obtained by our model (**a**) and the MDA model (**b**). The motion patterns are sorted in descending order according to the flow values.

### 5.3. Crowd Motion Description and Simulation

Crowd motion is the aggregation of pedestrian motions. The movement path of a pedestrian can be approximately represented by the shortest path from the origin to destination based on the shortest path principle. Therefore, we can describe the crowd motion with a few motion parameters, including the number of pedestrians; pedestrian entry time; origin and destination; and speed. The motion parameters are quite disparate in different time periods and different motion patterns. Thus, we divide the crowd motion

into several motion units and model the motion parameters of the motion units. A motion unit contains a motion pattern of the crowd motion in a time period. Here, each time period was set to 30 s. For each motion unit, we use the flow value of the observed trajectories to represent the number of pedestrians. Due to the short time period, we suppose that the entering times of pedestrians follow a uniform distribution. The origins and destinations of pedestrians are represented by a probability model based on the frequency of the trajectories reaching the grids at the entrance and exit, respectively. The speeds of pedestrians are represented by a normal distribution, and the mean and variance are calculated by the speeds of the trajectories. Next, we use the motion parameters and motion model to generate the virtual pedestrians, and their trajectories to simulate the crowd motion. The motion parameters are generated by the parameter models described above. To simulate the behavior independence of the pedestrians in crowd motion, we add random noise to the movement paths of virtual pedestrians. During the path generation of each pedestrian, we randomly increase the length of each path in the shortest path graph by $[0, 0.5]$ times.

To evaluate the validity of the crowd motion simulation, we propose a new indicator to measure the correlations between the simulated trajectories and the original trajectories obtained by the KLT tracker. A higher correlation means better effectiveness of the crowd motion simulation. We use the spatial occupancy values of trajectory to build the feature vector, and measure the similarity between the two groups of trajectories with the correlation between their feature vectors. The spatial occupancy value of trajectory refers to the appearance frequency of the trajectories within a certain spatial range in a time period. We divide the scene image into non-overlapping blocks with a size of $N \times N$ pixels. In a given time period, we calculate the sum of the number of trajectories appearing in a block at all time points and take it as the spatial occupancy value of the block. We arrange the spatial occupancy values of all image blocks into a feature vector according to the position of blocks, and then use the Pearson correlation coefficient to represent the correlation between the feature vectors. We set the sizes of image blocks to $60 \times 60$, $40 \times 40$ and $20 \times 20$ pixels; and calculated the correlations between the simulated trajectories and original trajectories in different time periods to evaluate the effectiveness of the crowd motion simulation. Each time period was set to 60 or 30 s. We show the correlation coefficients obtained by our model and the MDA model in Figure 10. The results are displayed separately according to the different sizes of blocks and different time periods. We can see that in most instances, the correlation coefficients obtained by our model are larger than those obtained by the MDA model, which means that our model can better simulate crowd motion in most cases. The average values of the correlation coefficients obtained by both models decreased as the block size and time period length were reduced. This is because that there must be differences between the simulated trajectories and original trajectories, which will become more obvious as the observation windows reduce. To intuitively illustrate the performance of our model in crowd motion simulations, we show three comparison groups of the real scene and our simulated scene with different crowd densities in Figure 11. We can see that in the three groups of scenes, the spatial distributions of the simulated pedestrians and real pedestrians are relatively close, which demonstrates the validity of our model from another perspective. There are, of course, some differences between the simulated scene and the real one due to the limitations of feature point extraction and the proposed model.

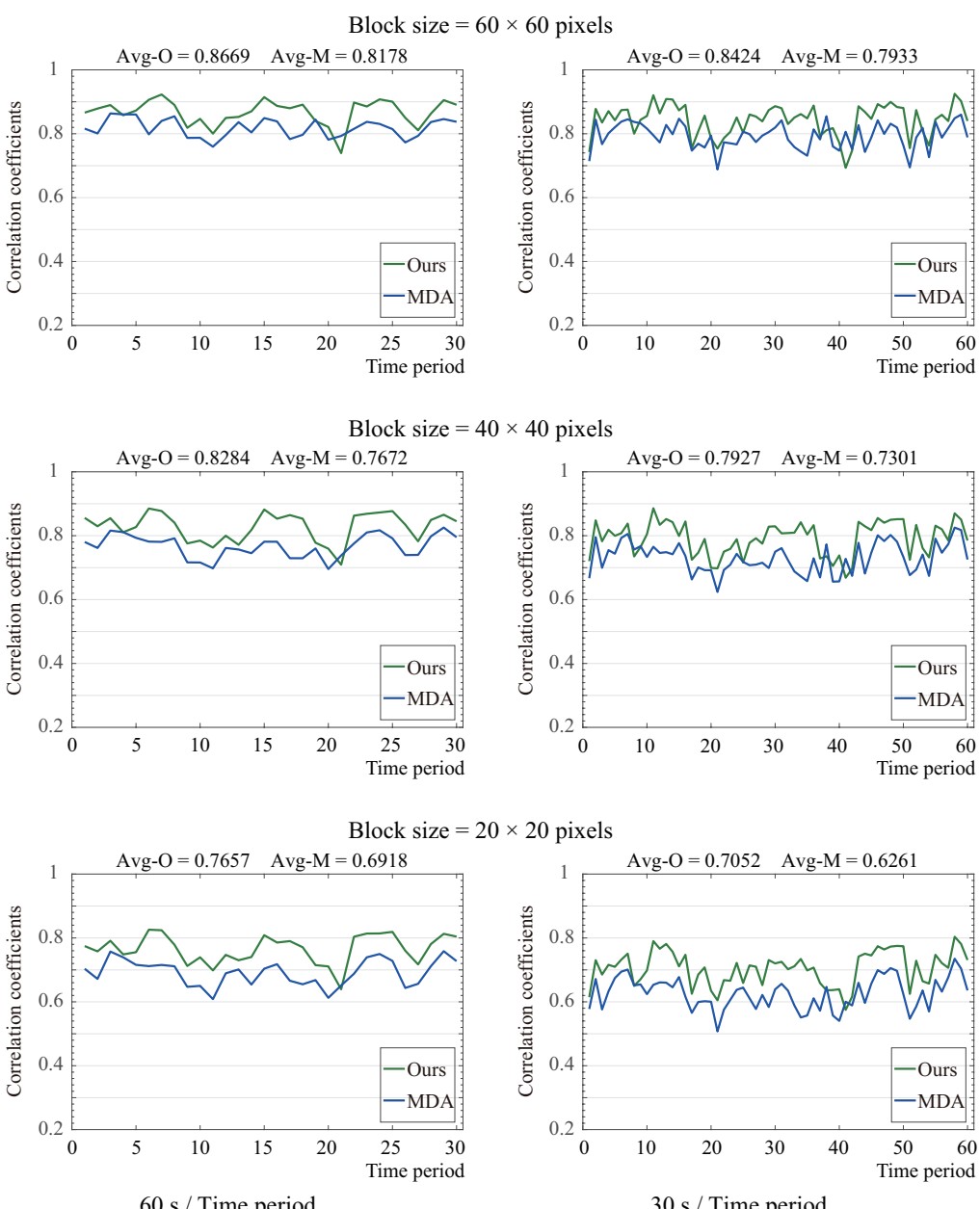

**Figure 10.** The correlation coefficients between the simulated trajectories and original trajectories obtained by our model and the MDA model. The average values of the correlation coefficients obtained by our model are larger than those obtained by the MDA model in all sub-figures. A larger correlation coefficient means better effectiveness of the crowd motion simulation. The *p*-values of the t-test for all correlation coefficients are less than 0.01.

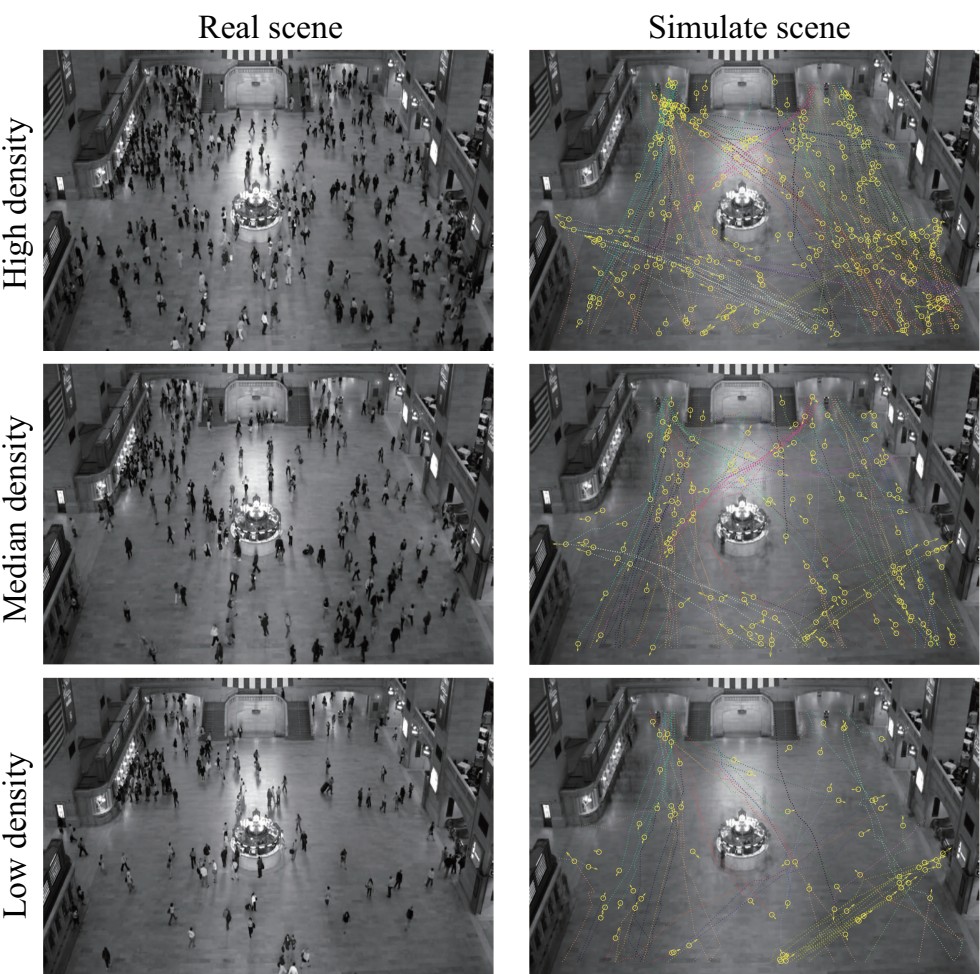

**Figure 11.** Three comparison groups of the real scene and our simulated scene with different crowd densities. The yellow small circles represent the simulated pedestrians. The colors of simulated trajectories were assigned according to the motion patterns.

## 6. Conclusions

In this work, we revisited the research results of sociologists to summarize the shortest path principle, and then proposed a new motion model to simulate the moving of pedestrians in crowded scenes. The proposed motion model consists of two parts: the shortest path graph and the generalized path likelihood. The shortest path graph generates the movement paths of pedestrians between pairs of points in a scene. The generalized path likelihood estimates the probability of a pedestrian reaching each candidate pair of origin and destination. We compared our motion model with the MDA model in experiments. The experimental results showed that our model outperforms the MDA model in pedestrian motion simulation, motion pattern classification and traffic flow statistics, and crowd motion description and simulation. In future work, the pedestrian motion model could be used as the prior knowledge for pedestrian tracking, a contrast template for pedestrian anomaly detection and so on.

**Author Contributions:** Conceptualization, Y.Z. and Y.L.; methodology, Y.Z.; software, Y.Z.; validation, Y.Z.; formal analysis, Y.Z.; investigation, Y.Z.; resources, Y.Z.; data curation, Y.Z.; writing—original draft preparation, Y.Z.; writing—review and editing, Y.Z. and Y.L.; visualization, Y.Z.; supervision, Y.L.; project administration, Y.L.; funding acquisition, Y.L. All authors have read and agreed to the published version of the manuscript.

**Funding:** This research received no external funding.

**Institutional Review Board Statement:** Not applicable.

**Informed Consent Statement:** Not applicable.

**Conflicts of Interest:** The authors declare no conflict of interest.

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
