# Peer review of "Modeling Pedestrian Motion in Crowded Scenes Based on the Shortest Path Principle"

_applsci, doi:10.3390/app12010381_

Round 1

Reviewer 1 Report

The authors of the manuscript provide a two part motion model for pedestrian  motion estimation.
The manuscript is well written and presented.

I am curious about the initiation method for  probability of the existence  as used in 2. Can the authors use a a markov model for probability of existence? such as in intergrated PDA family of algorithms?

Author Response

Response to the reviewer: Thanks for your comments and suggestions.

We guess the "probability of the existence" you mentioned is the probability of the existence of the generalized trajectory. In the formula, Str(ga,gb') represents the length of a feasible shortest path of the generalized trajectory, S(go,gb')-S(go,ga) represents the effective projection length of the generalized trajectory on the ideal shortest path from origin go to grid gb'. We used the ratio between them to measure the probability of the existence of the generalized trajectory. For a better expression, we added this explanation to the revised paper.

The probability of the existence of the generalized trajectory is only a component of the generalized path likelihood in this paper. The probability was not designed for tracking individuals in a crowd, so we did not use a complex probability model to describe it. Inspired by your suggestion, we will try to combine it with the Markov model and the probabilistic data association method in future work.

In addition, we checked the English expression and made some grammatical changes to the paper.

Reviewer 2 Report

The contribution that is reported in this paper is of a mixed kind. Thus, the theoretical algorithmic approach is relatively interesting. I particularly appreciate their consideration of Dynamic Pedestrian-Agents model, as a reference for their work. I also remark the interdisciplinary accents of this work, as they make psychological sociology-related considerations. Nevertheless, it is unclear how the described approach could be effectively applied in a particular use case scenario. The social psychology, as a key scientific field that intensively and extensively studied human collective behaviours, suggests that crowd behaviour pertains to certain rather well known rules and patterns, even the methods, which may be considered in order to influence crowd behaviour, are rather well known. Therefore, I would appreciate that the paper represents a rather theoretical-level approach, that may be used in order to eventually further enhance the understanding concerning human crowd behaviour. If the authors intend to utilize it in a more specific real-world use case, then this should be comprehensively described, including the used dataset and the considered real-world performance assessment methodology. Otherwise, the precise scope of the contribution should be clearly specified, as a mostly theoretical study concerning the algorithmics of human collective behaviour.

The authors should proofread and improve the English language and style throughout the paper.

Author Response

Response to the reviewer: Thanks for your comments and suggestions.

The objective of this paper is to describe crowd motion by computer vision technique. Inspired by the findings of social psychology, we proposed a pedestrian motion model based on the shortest path principle. We tested the performance of the proposed model for several common applications in the computer vision field. This paper is just a scientific attempt to fuse the domain knowledge from social psychology into computer vision techniques. Therefore, in this paper, we did not conduct in-depth theoretical research, nor did we design experiments according to the standards of practical application.

According to your comments, we made some modifications to specify the precise scope of the contribution.

“Abstract: In the computer vision field, understanding human dynamics is not only a great challenge but also a very meaningful work, which plays an indispensable role in public safety. Despite the complexity of human dynamics, physicists have found that the pedestrian motion in a crowd is governed by some internal rules, which can be formulated as a motion model, and an effective model is of great importance for understanding and reconstructing human dynamics in various scenes.”

“Keywords: Crowd motion, computer vision, motion model of pedestrians, shortest path principle, origin and destination”

In addition, we checked the English expression and made some grammatical changes to the paper.

Reviewer 3 Report

  1. Do you have a name for the new indicator proposed by you and a mathematical description of it? (Line 350)
  2. What do you mean by “the heat values of the trajectories’ spatial distribution”? Is this about the strengths of this distribution? (Line 353)
  3. Can you analyze numerically the results from Figure 10? Visually, it is obvious that your model has better results. Maybe you can specify the improvement made by your model by an average percentage value.
  4. I recommend a paper to be considered as a reference, that can be of interest for this subject: “State-of-the-art crowd motion simulation models” by Dorine C.Duives, WinnieDaamen1, Serge P.Hoogendoorn2,.

Author Response

Response to the reviewer: Thank you very much for the constructive comments and suggestions.

We have made several modifications to the paper according to your comments and suggestions. We responded to your comments one by one in the attached PDF and elaborated on the modifications.

In addition, we checked the English expression and made some grammatical changes to the paper.

Round 2

Reviewer 2 Report

The authors have finally acknowledged the fundamental drawback that I mentioned during my previous two review rounds, quote "This paper is just a scientific attempt to fuse the domain knowledge from social psychology into computer vision techniques. Therefore, in this paper, we did not conduct in-depth theoretical research, nor did we design experiments according to the standards of practical application." Consequently, if the paper is needed in this form(please also see my previous review comments), it can be accepted. The English language still has some issues and could be improved.

Author Response

Response to the reviewer: Thanks for your comments and suggestions.
According to your comments, our future work on crowd motion analysis will make efforts on the two aspects: deeper theoretical research and closer to the needs of practical applications.
In addition, we checked the English expression and made a few grammatical changes to the paper.